# Efficient Sparse Single-stage 3D Visual Grounding with Text-guided Pruning

## Abstract

In this paper, we propose an efficient sparse convolution-based architecture called ESS3D for 3D visual grounding. Conventional 3D visual grounding methods are difficult to meet the requirements of real-time inference due to the two-stage or point-based architecture. Inspired by the success of multi-level fully sparse convolutional architecture in 3D object detection, we aim to build a new 3D visual grounding framework following this technical route. However, as in visual grounding task the 3D scene representation should be deeply interacted with text features, sparse convolution-based architecture is inefficient for this interaction due to the large amount of voxel features. To this end, we propose text-guided pruning (TGP) and completion-based addition (CBA) to deeply fuse 3D scene representation and text features in an efficient way by gradual region pruning and target completion. Specifically, TGP iteratively sparsifies the 3D scene representation and thus efficiently interacts the voxel features with text features by cross-attention. To mitigate the affect of pruning on delicate geometric information, CBA adaptively fixes the over-pruned region by voxel completion with negligible computational overhead. Compared with previous single-stage methods, ESS3D achieves top inference speed and surpasses previous fastest method by 100% FPS. ESS3D also achieves state-of-the-art accuracy even compared with two-stage methods, with +1.13 lead of Acc@0.5 on ScanRefer, and +5.4 and +5.0 leads on NR3D and SR3D respectively. The code will be released soon.

## 1 Introduction

Incorporating multi-modal information to guide 3D visual perception is a promising direction. In these years, 3D visual grounding (3DVG), also known as 3D instance referencing, has been paid increasing attention as a fundamental multi-modal 3D perception task. The aim of 3DVG is to locate an instance in the scene using 3D bounding box, where this instance is described by a free-form query language. 3DVG is challenging since it requires understanding of both 3D scene and language description. Recently, with the development of 3D scene perception and vision-language models, 3DVG methods have shown remarkable progress (Jain et al., 2022; Luo et al., 2022). However, with 3DVG being widely applied in fields like robotics and AR / VR where inference speed is the main bottleneck, how to construct efficient real-time 3DVG model remains a challenging problem.

Since the output format of 3DVG is similar with 3D object detection, early 3DVG methods (Yuan et al., 2021; Yang et al., 2021) usually adopt a two-stage framework, which first conducts 3D object detection to locate all objects in a scene, and then selects the target object by incorporating text information. As there are many similarities between 3D object detection and 3DVG (e.g. both of them need to extract the representation of the 3D scene), there will be much redundant feature computation during the independent adoption of the two models. As a result, two-stage methods are usually slow and hard to handle real-time tasks. To solve this problem, single-stage methods (Luo et al., 2022) are presented, which generates the bounding box of the target object directly from point clouds. This integrated design makes single-stage methods more compact and efficient. However, current single-stage 3DVG methods mainly build on point-based architecture (Qi et al., 2017), where the feature extraction contains time-consuming operations like furthest point sampling and kNN, especially for large scenes. They also need to aggressively downsample the point features to reduce computational cost, which might hurt the geometric information of small and thin objects (Xu et al.,

2024). Due to these reasons, current single-stage methods are still far from real-time ($< 6$ FPS) and their performance is inferior to two-stage methods.

In this paper, we propose a new single-stage framework for 3DVG based on sparse convolution, namely ESS3D. Inspired by state-of-the-art 3D object detection methods (Rukhovich et al., 2022; Xu et al., 2024) which achieves both leading accuracy and speed with multi-level sparse convolutional architecture, we build the first sparse single-stage 3DVG network. However, different from 3D object detection, in 3DVG the 3D scene representation should be deeply interacted with text features. Since the amount of voxels is very large in sparse convolution-based architecture, deep multi-modal interaction like cross-attention becomes infeasible due to unaffordable computational cost. To this end, we propose text-guided pruning (TGP), which first utilize text information to jointly sparsify the 3D scene representation and enhance the voxel and text features. To mitigate the affect of pruning on delicate geometric information, we further present completion-based addition (CBA) to adaptively fix the over-pruned region with negligible computational overhead. Specifically, TGP prunes the voxel features according to the object distribution. It gradually removes background features and features of irrelevant objects through iterative pruning and feature upsampling, which generates high-resolution and text-aware voxel features around the target object for accurate bounding box prediction. Since pruning may mistakenly remove the representation of target object, CBA utilizes text features to query a small set of voxel features from the complete backbone features, followed by pruned-aware addition and voxel concatenation to fix the over-pruned region. We conduct extensive experiments on the popular ScanRefer (Chen et al., 2020) and ReferIt3D (Achlioptas et al., 2020) datasets. Compared with previous single-stage methods, ESS3D achieves top inference speed and surpasses previous fastest single-stage method by 100% FPS. ESS3D also achieves state-of-the-art accuracy even compared with two-stage methods, with $+1.13$ lead of Acc@0.5 on ScanRefer, and $+5.4$ and $+5.0$ leads on NR3D and SR3D respectively.

## 2 RELATED WORK

### 2.1 3D VISUAL GROUNDING

3D visual grounding aims to locate a target object within a 3D scene based on natural language descriptions. Existing methods are typically categorized into two-stage and single-stage approaches. Two-stage methods follow a detect-then-match paradigm. In the first stage, they independently extract features from the language query using pre-trained language models (Devlin, 2018; Pennington et al., 2014) and predict candidate 3D objects using pre-trained 3D detectors (Qi et al., 2019; Liu et al., 2021). In the second stage, they focus on aligning the vision and text features to identify the target object. Techniques for feature fusion include attention mechanisms with Transformers (He et al., 2021; Zhao et al., 2021), contrastive learning (Abdelreheem et al., 2022), and graph-based matching (Feng et al., 2021). Single-stage methods integrate object detection and feature extraction, allowing for direct identification of the target object. Methods in this category include guiding keypoint selection using textual features (Luo et al., 2022), and measuring similarity between words and objects inspired by 2D image-language pre-trained models like GLIP (Li et al., 2022), as in BUTD-DETR (Jain et al., 2022). And methods like EDA (Wu et al., 2023) and $G^3$-LQ (Wang et al., 2024) advance single-stage 3D visual grounding by enhancing multimodal feature discriminability through explicit text-decoupling, dense alignment, and semantic-geometric modeling.

However, existing two-stage and single-stage methods generally have high computational costs, hindering real-time applications. Our work aims to address these efficiency challenges by proposing an efficient single-stage method with multi-level sparse convolutional architecture.

### 2.2 MULTI-LEVEL SPARSE CONVOLUTIONAL ARCHITECTURES

Recently, multi-level sparse convolutional architecture has achieved great success in the field of 3D object detection. Built on the voxel-based representation (Wang et al., 2022) and sparse convolution operation (Choy et al., 2019; Graham et al., 2018; Xu et al., 2023), this kind of methods show great efficiency and accuracy when processing scene-level 3D data. GSDN (Gwak et al., 2020) first adopts sparse convolution in 3D object detection by constructing multi-level architecture with generative feature upsampling. FCAF3D (Rukhovich et al., 2022) simplifies the multi-level architecture with anchor-free design and rotation-aware object assignment strategy, which achieves leading accuracy

with even faster speed. Aimed at real-time 3D object detection, TR3D (Rukhovich et al., 2023) further accelerates FCAF3D by removing unnecessary layers and introducing category-aware proposal assignment method. Additionally, DSPDet3D Xu et al. (2024) introduces the multi-level architecture to 3D small object detection and demonstrates great accuracy and efficiency, even being able to process building-level 3D scenes.

Our proposed method draws inspiration from these approaches, utilizing a sparse multi-level architecture with sparse convolutions and an anchor-free design. This allows for efficient processing of 3D data, enabling real-time performance in 3D visual grounding tasks.

## 3 METHOD

In this section, we describe our ESS3D for efficient single-stage 3DVG. We first analyze existing pipelines to identify current challenges and motivate our approach (Sec. 3.1). We then introduce the text-guided pruning, which leverages text features to guide feature pruning (Sec. 3.2). To address the potential risk of pruning key information, we propose the completion-based addition for multi-level feature fusion (Sec. 3.3). Finally, we detail the training loss (Sec. 3.4).

### 3.1 MULTI-LEVEL SPARSE CONVOLUTIONAL ARCHITECTURE

Top-performance 3DVG methods (Wang et al., 2024; Wu et al., 2023; Shi et al., 2024), are mainly two-stage, which is a serial combination of 3D object detection and 3D object grounding. This separate calls of two approaches result in redundant feature extraction and complex pipeline, thus making the two-stage methods less efficient. To demonstrate the efficiency of the two-stage methods, we conduct a comparison of accuracy and speed among several representative methods on ScanRefer (Chen et al., 2020), as shown in Fig. 1. It can be seen that two-stage methods struggle in speed ($<$ 3 FPS) due to the additional detection stage. Since 3D visual grounding is usually adopted in practical scenarios that require real-time inference under limited resources, such as embodied robots and VR/AR, the low speed of two-stage methods make them less practical in real applications. On the other side, single-stage methods (Luo et al., 2022), which directly predicts refered bounding box from the observed 3D scene, are more suitable choices due to their streamlined processes. We also list the accuracy-speed tradeoff of single-stage methods in Fig. 1. It is shown that they are much more efficient than the two-stage counterparts.

However, existing single-stage methods are mainly built on point-based backbone (Qi et al., 2017), where the scene representation is extracted with time-consuming operations like furthest point sampling and set abstraction. They also employ large transformer decoder to fuse text and 3D features for several iterations. Therefore, the inference speed of current single-stage methods is still far from real-time ($<$ 6 FPS). Inspired by the success of single-stage fully sparse convolutional architecture in the field of 3D object detection (Rukhovich et al., 2023), which achieves both leading accuracy and speed, we propose to build the first sparse convolution-based single-stage 3DVG pipeline.

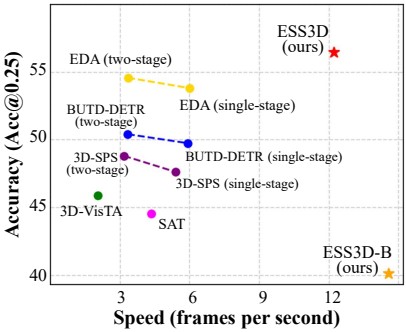

Figure 1: Comparison of state-of-the-art 3DVG methods on ScanRefer.

**ESS3D-B.** Here we propose a baseline framework based on sparse convolution, namely ESS3D-B. Following the simple and effective multi-level architecture of FCAF3D (Rukhovich et al., 2022), ESS3D-B utilizes 3 levels of sparse convolutional blocks for scene representation extraction and bounding box prediction, as shown in Fig. 2 (a). Specifically, the input pointclouds $P \in \mathbb{R}^{N \times 6}$ with 6-dim features (3D position and RGB) are first voxelized and then fed into three sequential MinkResBlocks (Choy et al., 2019) with stride 2, which generates three levels of voxel features $V_l$ ($l = 1, 2, 3$). With the increase of $l$, the spatial resolution of $V_l$ decreases and the context information increases. Concurrently, the free-form text with $l$ words is encoded by the pre-trained RoBERTa (Liu, 2019) and produce the vanilla text tokens $T \in \mathbb{R}^{l \times d}$. With the extracted 3D and text representations, we iteratively upsample $V_3$ and fuse it with $T$ to

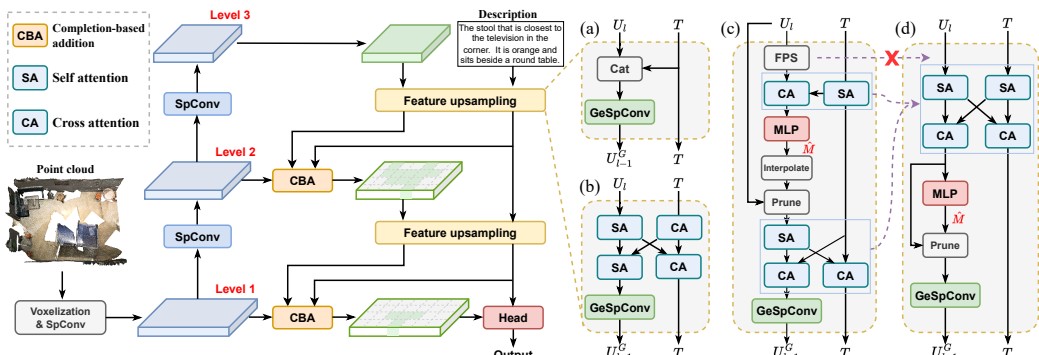

Figure 2: Illustration of ESS3D. ESS3D builds on multi-level sparse convolutional architecture. It iteratively upsamples the voxel features with text-guided pruning (TGP), and fuses multi-level features via completion-based addition (CBA). (a) to (d) on the right side illustrate various options for feature upsampling. (a) refers to simple concatenation with text features, which is fast but less accurate. (b) refers to feature interaction through cross-modal attention mechanisms, which is constrained by the large amount of voxels. (c) represents our proposed TGP, which first prunes voxel features under textual guidance and thus enables efficient interaction between voxel and text features. (d) shows a simplified version of TGP that removes feature sampling and interpolation, combines multi-modal feature interactions into a whole and moves it before pruning.

generate high-resolution and text-aware scene representation:

$$U_l = U_l^G + V_l, \quad U_l^G = \text{GeSpConv}(U_{l+1}'), \quad U_{l+1}' = \text{Concat}(U_{l+1}, T) \tag{1}$$

where $U_3 = V_3$, GeSpConv means generative sparse convolution (Gwak et al., 2020) with stride 2, which upsamples the voxel features and expands their spatial locations for better bounding box prediction. Concat is voxel-wise feature concatenation by duplicating $T$. The final upsampled feature map $U_1$ is concatenated with $T$ and fed into a convolutional head to predict the objectness scores and regress the 3D bounding box, where each voxel feature is regarded as an object proposal and used to predict box. We select the box with highest objectness score as the grounding result.

As shown in Fig. 1, ESS3D-B achieves an inference speed of 14.58 FPS, which is significantly faster than previous single-stage methods and demonstrates great potential for real-time 3DVG.

### 3.2 TEXT-GUIDED PRUNING

Though efficient, ESS3D-B exhibits poor performance due to the inadequate interaction between 3D scene representation and text features. Motivated by previous 3DVG methods (Jain et al., 2022), a simple solution is to replace Concat with cross-modal attention to update voxel and text features with two intertwined transformer decoders that process them jointly via cross-attention, as shown in Fig. 2 (b). However, different from point-based architectures where the scene representation is usually aggressively downsampled to control the computational cost, the amount of voxels in multi-level sparse convolutional framework is very large. In practical implementation, we find that the voxels expand almost exponentially with each upsampling layer, leading to a substantial computational burden for the self-attention and cross-attention of scene features. To address this issue, we introduce text-guided pruning (TGP) to construct ESS3D, as illustrated in Fig. 2 (c). The core idea of TGP is to reduce feature size by pruning redundant voxels and guide the network to gradually focus on the final target based on textual features.

**Overall Architecture.** TGP can be regarded as a modified version of cross-modal attention, which reduces the amount of voxels before attention operation to reduce the computational cost. To minimize the affect of pruning on the final prediction, we propose to prune the scene representation gradually. At higher level where the amount of voxels is not too large yet, TGP prunes less voxels. While at lower level where the amount of voxels is significantly increased by upsampling operation, TGP prunes the voxel features more aggressively. The multi-level architecture of ESS3D consists of three levels and includes two feature upsampling operations. Therefore, we correspondingly configure two TGPs with different functions, which are referred as scene-level TGP (level 3 to 2) and

target-level TGP (level 2 to 1) respectively. Scene-level TGP aims to distinguish between objects and the background within the scene, specifically pruning the voxels on background. Target-level TGP focuses on regions mentioned in the text, intending to preserve the target object and referential objects while removing other regions.

**Details of TGP.** Since the pruning is relevant to the description, we need to make the voxel features text-aware to predict a proper pruning mask. To reduce the computational cost, we perform farthest point sampling (FPS) on the voxel features to reduce their size while preserving the basic distribution of the scene. Next, we utilize cross-attention to interact with the text features and employ a simple MLP to predict the probability distribution $\hat{M}$ for retaining each voxel. To prune the features $U_l$, we binarize and interpolate the $\hat{M}$ to obtain the pruned mask. This process can be expressed as:

$$U_l^P = U_l \odot \Theta(\mathcal{I}(\hat{M}, U_l) - \sigma), \quad \hat{M} = \text{MLP}(\text{CrossAtt}(\text{FPS}(U_l), \text{SelfAtt}(T))) \quad (2)$$

where $U_l^P$ is the pruned features, $\Theta$ is Heaviside step function, $\odot$ is matrix dot product, $\sigma$ is the pruning threshold, and $\mathcal{I}$ represents linear interpolation based on the positions specified by $U_l$. After pruning, the scale of the scene features is significantly reduced, enabling internal feature interactions based on self-attention. Subsequently, we utilize self-attention and cross-attention to perceive the relative relationships among objects within the scene and to fuse multimodal features, resulting in updated features $U_l'$. Finally, through generative sparse convolutions, we obtain $U_{l-1}^G$.

**Supervision for Pruning.** The binary supervision mask $M^{sce}$ for scene-level TGP is generated based on the centers of all objects in the scene, and the mask $M^{tar}$ for target-level TGP is based on the target and relevant objects mentioned in the descriptions:

$$M^{sce} = \bigcup_{i=1}^{N} \mathcal{M}(O_i), \quad M^{tar} = \mathcal{M}(O^{tar}) \cup \bigcup_{j=1}^{K} \mathcal{M}(O_j^{rel}) \quad (3)$$

where $\{O_i | 1 \le i \le N\}$ indicates all objects in the scene. $O^{tar}$ and $O^{rel}$ refer to target and relevant objects respectively. $\mathcal{M}(O)$ represents the mask generated from the center of object $O$. It generates a $L \times L \times L$ cube centered at the center of $O$ to construct the supervision mask $M$, where locations inside the cube is set to 1 while others set to 0.

**Simplification.** Although the above mentioned method can effectively prune the voxel features based on text description to reduce the computational cost of cross-modal attention, there are some inefficient operations in the pipeline: (1) FPS is time-consuming, especially for large scenes; (2) there are two times of interactions between voxel features and text features, the first is to guide pruning and the second is to enhance the representation, which is a bit redundant. We also empirically observe that the amount of voxels is not large in level 3. To this end, we propose a simplified version of TGP, as shown in Fig. 2 (d). We remove the FPS and merge the two multi-modal interactions into one. We also move the merged interaction operation before pruning. In this way, voxel features and text features are first deeply interacted for both feature enhancement and pruning. Because in level 3 the amount of voxels is small and in level 2 / 1 the voxels are already pruned, the computational cost of self-attention and cross-attention is always kept at a relatively low level.

**Effectiveness of TGP.** After pruning, the voxel scale of $U_1$ is reduced to nearly 15% of its original size without TGP, while the 3DVG performance is significantly boosted. TGP serves multiple functions, including: (1) facilitating the interaction of multi-modal features through cross-attention, (2) reducing the feature scale (amount of voxels) through pruning, and (3) gradually guiding the network to focus on the mentioned target based on text features.

### 3.3 COMPLETION-BASED ADDITION

During the pruning process, some targets may be mistakenly removed, especially small or narrow objects, as shown in Fig. 3 (b). Therefore, the addition operation between the upsampled pruned features $U_l^G$ and backbone features $V_l$ described in Equation (1) play an important role to mitigate the affect of over-pruning.

There are two alternative addition operation: (1) **Full Addition.** For the intersecting regions of $V_l$ and $U_l^G$, features are directly added. For voxel features outside the intersection of $U_l^G$ and $V_l$ which lack corresponding features in the other map, the missing voxel features are interpolated from the

other before addition. Due to pruning process, $U_l^G$ is sparser than $V_l$. In this way, full addition can fix almost all the pruned region to avoid any risk. But this operation is computationally heavy and cannot make the scene representation focus on relevant objects, which deviates the core idea of TGP. (2) **Pruning-aware Addition.** The addition is constrained to the locations of $U_l^G$. For voxel in $U_l^G$ but not in $V_l$, interpolation from $U_l^G$ is applied to complete the missing locations in $V_l$. It restricts the addition operation to the shape of the pruned features, potentially leading to an over-reliance on the results of the pruning process. If some important regions are over-pruned, the network might struggle to detect small or narrow targets whose geometric information is seriously damaged.

Considering the unavoidable risk of pruning the query target or important relevant objects, we introduce the completion-based addition (CBA). CBA is designed to mitigate the drawback of both full and pruning-aware addition by providing a more targeted and efficient way of integrating multi-level features, ensuring that essential details are preserved during the feature fusion process while the additional computational overhead is negligible.

**Details of CBA.** We first enhance the backbone features $V_l$ with the text features $T$ through cross-attention, obtaining $V_l'$. Then a MLP is adopted to predict the probability distribution of target for region selection:

$$M_l^{tar} = \Theta(\mathrm{MLP}(V_l') - \tau) \qquad (4)$$

where $\Theta$ is the step function, and $\tau$ is the threshold determining voxel relevance. $M_l^{tar}$ is a binary mask indicating potential regions of the mentioned target. Then, comparison of $M_l^{tar}$ with $U_l$ identifies missing voxels. The missing mask $M_l^{mis}$ is derived as follows:

$$M_l^{mis} = M_l^{tar} \wedge (\neg\, \mathcal{C}(U_l^G, V_l)) \qquad (5)$$

where $\mathcal{C}(A, B)$ denotes the generation of a binary mask for $A$ based on the shape of $B$. For positions in $B$, if there are corresponding voxel features in $A$, the mask for that position is set to 1. Otherwise it is set to 0. Missed voxel features in $U_l^G$ that correspond to $M_l^{mis}$ are interpolated from $U_l^G$, filling in gaps identified by the missing mask. The completed feature map $U_l^{cpl}$ is computed by:

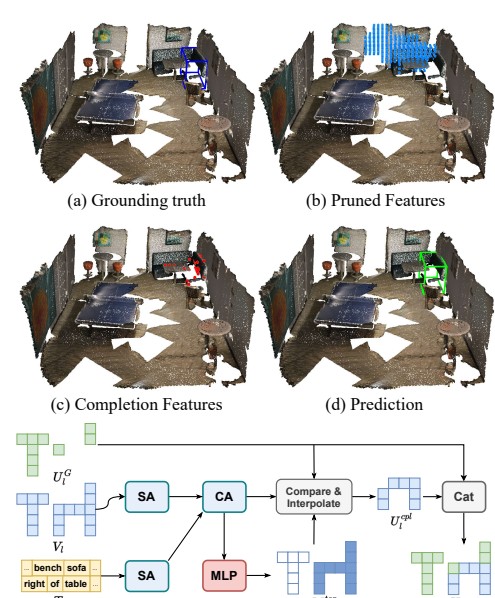

(a) Grounding truth  (b) Pruned Features

(c) Completion Features  (d) Prediction

Figure 3: Illustration of completion-based addition. The above (b) illustrate an example of over-pruning on the target. (c) refers to the completed features predicted by CBA. The lower diagram demonstrates how CBA predicts target distribution under textual guidance and adaptively completes the pruned features.

$$U_l^{cpl} = V_l' \odot M_l^{mis} + \mathcal{I}(U_l^G, M_l^{mis}) \qquad (6)$$

where $\mathcal{I}$ represents linear interpolation on the feature map based on the positions specified in the mask. Finally, the original upsampled features are combined with the backbone features according to the pruning-aware addition, and merged with the completion features to yield the updated $U_l$:

$$U_l = \mathrm{Concat}(U_l^G \leftarrow V_l, U_l^{cpl}) \qquad (7)$$

where $\leftarrow$ denotes the pruning-aware addition, and Concat means concatenation of voxel features.

## 3.4 Train Loss

The loss of ESS3D is composed of several components: pruning loss for TGP, completion loss for CBA, and objectness loss as well as bounding box regression loss for the head. Pruning loss, completion loss and objectness loss employ the focal loss to handle class imbalance. Supervision for completion and classification losses are the same, which sets voxels near the target object center as positives while leaving others as negatives. For bounding box regression, we use the Distance-IoU (DIoU) loss. The total loss function is computetd as the sum of these individual losses:

$$\mathcal{L}_{\text{total}} = \lambda_1 \mathcal{L}_{\text{pruning}} + \lambda_2 \mathcal{L}_{\text{com}} + \lambda_3 \mathcal{L}_{\text{class}} + \lambda_4 \mathcal{L}_{\text{bbox}}$$

where $\lambda_1$, $\lambda_2$, $\lambda_3$ and $\lambda_4$ are the weights of different parts.

Table 1: Comparison of methods on the ScanRefer dataset evaluated at IoU thresholds of 0.25 and 0.5. ESS3D achieves state-of-the-art accuracy even compared with two-stage methods, with +1.13 lead on Acc@0.5. Notably, we are the first to comprehensively evaluate inference speed for 3DVG methods. The inference speeds of other methods are obtained through our reproduction.

| Method | Venue | Pipeline | Input | Accuracy 0.25 | 0.5 | Inference Speed (FPS) |
|---|---|---|---|---|---|---|
| ScanRefer (Chen et al., 2020) | ECCV'20 | Two-stage | 3D+2D | 41.19 | 27.40 | 6.72 |
| TGNN (Huang et al., 2021) | AAAI'21 | Two-stage | 3D | 37.37 | 29.70 | 3.19 |
| InstanceRefer (Yuan et al., 2021) | ICCV'21 | Two-stage | 3D | 40.23 | 30.15 | 2.33 |
| SAT (Yang et al., 2021) | ICCV'21 | Two-stage | 3D+2D | 44.54 | 30.14 | 4.34 |
| FFL-3DOG (Feng et al., 2021) | ICCV'21 | Two-stage | 3D | 41.33 | 34.01 | Not release |
| 3D-SPS (Luo et al., 2022) | CVPR'22 | Two-stage | 3D+2D | 48.82 | 36.98 | 3.17 |
| BUTD-DETR (Jain et al., 2022) | ECCV'22 | Two-stage | 3D | 50.42 | 38.60 | 3.33 |
| EDA (Wu et al., 2023) | CVPR'23 | Two-stage | 3D | 54.59 | 42.26 | 3.34 |
| 3D-VisTA (Zhu et al., 2023) | ICCV'23 | Two-stage | 3D | 45.90 | 41.50 | 2.03 |
| VPP-Net (Shi et al., 2024) | CVPR'24 | Two-stage | 3D | 55.65 | 43.29 | Not release |
| $G^3$-LQ (Wang et al., 2024) | CVPR'24 | Two-stage | 3D | **56.90** | 45.58 | Not release |
| 3D-SPS (Luo et al., 2022) | CVPR'22 | Single-stage | 3D | 47.65 | 36.43 | 5.38 |
| BUTD-DETR (Jain et al., 2022) | ECCV'22 | Single-stage | 3D | 50.22 | 37.87 | 5.91 |
| EDA (Wu et al., 2023) | CVPR'23 | Single-stage | 3D | 53.83 | 41.70 | 5.98 |
| $G^3$-LQ (Wang et al., 2024) | CVPR'24 | Single-stage | 3D | 55.95 | 44.72 | Not release |
| ESS3D (Ours) | —— | Single-stage | 3D | 56.45 | **46.71** | **12.43** |

## 4 EXPERIMENTS

### 4.1 DATASETS

We maintain the same experimental settings with previous works, employing ScanRefer (Chen et al., 2020) and SR3D/NR3D (Achlioptas et al., 2020) as datasets. **ScanRefer**: Built on the ScanNet framework, ScanRefer includes 51,583 descriptions across scenes, with an average of 13.81 objects per scene. Evaluation metrics focus on Acc@$m$IoU, categorizing predictions into "unique" and "multiple" based on object singularity within the scene. **ReferIt3D**: Also based on ScanNet, this dataset splits into Nr3D, with 41,503 human-generated descriptions, and Sr3D, containing 83,572 synthetic expressions. ReferIt3D simplifies the task by providing segmented point clouds for each object, requiring only classification and selection of target objects. The primary evaluation metric is accuracy in target object selection.

### 4.2 IMPLEMENTATION DETAILS

ESS3D is implemented using PyTorch and MinkowskiEngine. The pruning thresholds are set at $\sigma_{\text{sce}} = 0.7$ and $\sigma_{\text{tar}} = 0.3$, and the completion threshold in CBA is $\tau = 0.15$. The initial voxelization of the point cloud has a voxel size of 1cm, while the voxel size for level $i$ features scales to $2^{i+2}$ cm. The supervision for pruning uses $L = 7$. The weights for all components of the loss function, $\lambda_1, \lambda_2, \lambda_3, \lambda_4$, are equal to 1. Training is conducted using four GPUs, while inference speeds are evaluated using a single consumer-grade GPU, RTX 3090, with a batch size of 1.

### 4.3 QUANTITATIVE COMPARISONS

**Performance on ScanRefer.** We carry out comparisons with SOTA methods on ScanRefer dataset, as detailed in Tab. 1. The inference speeds of other methods are obtained through our reproduction with a single RTX 3090 and a batch size of 1. For two-stage methods, the inference speed includes the time taken for object detection in the first stage. For methods using 2D image features and 3D point clouds as inputs, we do not account for the time spent extracting 2D features, assuming they can be obtained in advance. However, in practical applications, the acquisition of 2D features also impacts overall efficiency. ESS3D achieves state-of-the-art accuracy even compared with two-stage methods, with +1.13 lead on Acc@0.5. Notably, in the single-stage setting, ESS3D

Table 2: Quantitative comparisons on Nr3D and Sr3D dataset. We evaluate under three pipelines, noting that the Two-stage using Ground-Truth Boxes is impractical for real-world applications. ESS3D exhibits significant superiority, with leads of $+5.4\%$ and $+5.0\%$ on NR3D and SR3D.

| Method | Venue | Pipeline | Accuracy | |
|---|---|---|---|---|
| | | | Nr3D | Sr3D |
| InstanceRefer (Yuan et al., 2021) | ICCV'21 | Two-stage (gt) | 38.8 | 48.0 |
| LanguageRefer (Roh et al., 2022) | CoRL'22 | Two-stage (gt) | 43.9 | 56.0 |
| 3D-SPS (Luo et al., 2022) | CVPR'22 | Two-stage (gt) | 51.5 | 62.6 |
| MVT (Huang et al., 2022) | CVPR'22 | Two-stage (gt) | 55.1 | 64.5 |
| BUTD-DETR (Jain et al., 2022) | ECCV'22 | Two-stage (gt) | 54.6 | 67.0 |
| EDA (Wu et al., 2023) | CVPR'23 | Two-stage (gt) | 52.1 | 68.1 |
| VPP-Net (Shi et al., 2024) | CVPR'24 | Two-stage (gt) | 56.9 | 68.7 |
| $G^3$-LQ (Wang et al., 2024) | CVPR'24 | Two-stage (gt) | 58.4 | 73.1 |
| InstanceRefer (Yuan et al., 2021) | ICCV'21 | Two-stage (det) | 29.9 | 31.5 |
| LanguageRefer (Roh et al., 2022) | CoRL'22 | Two-stage (det) | 28.6 | 39.5 |
| BUTD-DETR (Jain et al., 2022) | ECCV'22 | Two-stage (det) | 43.3 | 52.1 |
| EDA (Wu et al., 2023) | CVPR'23 | Two-stage (det) | 40.7 | 49.9 |
| 3D-SPS (Luo et al., 2022) | CVPR'22 | Single-stage | 39.2 | 47.1 |
| BUTD-DETR (Jain et al., 2022) | ECCV'22 | Single-stage | 38.7 | 50.1 |
| EDA (Wu et al., 2023) | CVPR'23 | Single-stage | 40.0 | 49.7 |
| ESS3D | —— | Single-stage | **48.7** | **57.1** |

achieves real-time performance, which is unprecedented among the existing methods. This significant improvement is attributed to our method's efficient use of a multi-level architecture based on 3D sparse convolutions, coupled with the text-guided pruning. By focusing computation only on salient regions of the point clouds, determined by textual cues, our model effectively reduces computational overhead while maintaining high accuracy. This enables our system to provide a viable solution for real-time efficient 3D visual grounding. ESS3D also sets a benchmark for inference speed comparisons for future methodologies.

**Performance on Nr3D/Sr3D.** We evaluate our method on the SR3D and NR3D datasets, following the evaluation protocols of prior works like EDA (Wu et al., 2023) and BUTD-DETR (Jain et al., 2022) by using Acc@0.25 as the primary accuracy metric. The results are shown in Tab. 2. Given that SR3D and NR3D provide ground-truth boxes and categories for all objects within a scene, we consider three different pipelines for evaluation: (1) Two-stage using Ground-Truth Boxes, (2) Two-stage using Detected Boxes, and (3) Single-stage. In practical applications, the Two-stage using Ground-Truth Boxes pipeline is unrealistic because obtaining all ground-truth boxes in a scene is infeasible. This approach can also oversimplify certain evaluation scenarios, rendering them less meaningful. For example, if there are no other objects of the same category as the target in the scene, the task reduces to relying on the provided ground-truth category. Under the Single-stage setting, we achieve peak performance of $48.7\%$ and $57.1\%$ on Nr3D and Sr3D. ESS3D exhibits significant superiority, even outperforming previous works under the pipeline of Two-stage using Detected Boxes, with leads of $+5.4\%$ and $+5.0\%$ on NR3D and SR3D datasets.

## 4.4 Ablation Study

**Effectiveness of Proposed Components.** To investigate the effects of our proposed TGP and CBA, we conduct ablation experiments with module removal as shown in the Tab. 3. When TGP is not used, multi-modal feature concatenation is employed as a replacement, as shown in Fig. 2 (a). When CBA is not used, it is substituted with a pruning-based addition. The results demonstrate that TGP significantly enhances performance without notably impacting inference time. This is because TGP, while utilizing a more complex multi-modal attention mechanism for stronger feature fusion, significantly reduces feature scale through text-guided pruning. Additionally, the performance improvement is also due to the gradual guidance towards the target object by both scene-level and target-level TGP. Implementing CBA on top of TGP further enhances performance, as CBA dynamically compensates for some of the excessive pruning by TGP, thus increasing the network's robustness.

Table 3: Effectiveness of the proposed TGP and CBA. Evaluated on the ScanRefer dataset.

| ID | TGP | CBA | Accuracy 0.25 | 0.5 | Speed (FPS) |
|---|---|---|---|---|---|
| (a) | | | 40.13 | 32.87 | **14.58** |
| (b) | ✓ | | 55.20 | 46.15 | 13.22 |
| (c) | | ✓ | 41.34 | 33.09 | 13.51 |
| (d) | ✓ | ✓ | **56.45** | **46.71** | 12.43 |

Table 4: Influence of the two CBAs at different levels. Evaluated on the ScanRefer dataset.

| ID | CBA (level 1) | CBA (level 2) | Accuracy 0.25 | 0.5 | Speed (FPS) |
|---|---|---|---|---|---|
| (a) | | | 55.20 | 46.15 | **13.22** |
| (b) | ✓ | | 55.17 | 46.06 | 12.79 |
| (c) | | ✓ | **56.45** | **46.71** | 12.43 |
| (d) | ✓ | ✓ | 56.22 | 46.68 | 12.19 |

**Integration of the Two CBAs.** To explore the impact of CBAs at two different levels, we conduct ablation experiments as depicted in Tab. 4. In the absence of CBA, we use pruning-based addition as a substitute. The results indicate that the CBA at level 2 has negligible effects on the 3DVG task. This is primarily because the CBA at level 2 mainly serves to supplement the scene-level TGP, which is tasked with pruning the background—a relatively straightforward process. Moreover, although some target features are pruned, they are compensated by two subsequent generative sparse convolutions. However, the CBA at level 1 enhances performance by adapt completion for the target-level TGP. It is challenging for the target-level TGP to fully preserve target objects through upsampling features, especially for smaller or narrower targets. The CBA at level 1, based on high-resolution backbone features, effectively complements the TGP.

**Feature Upsampling Techniques.** We conduct ablation experiments to assess the effects of different feature upsampling techniques, as detailed Tab. 5. As depicted in Fig. 2 (a), using simple feature concatenation, while fast in inference speed, results in poor performance. When we attempt to utilize an attention mechanism with stronger feature interaction capabilities, as shown in Fig. 2 (b), the computation exceeds the limits of GPU due to the

Table 5: Influence of different feature upsampling methods. Evaluated on the ScanRefer dataset.

| ID | Method | Accuracy 0.25 | 0.5 | Speed (FPS) |
|---|---|---|---|---|
| (a) | Simple concatenation | 40.13 | 32.87 | **14.58** |
| (b) | Attention mechanism | — | — | — |
| (c) | Text-guided pruning | 56.27 | 46.58 | 10.11 |
| (d) | Simplified TGP | **56.45** | **46.71** | 12.43 |

large number of voxels, making it impractical for real-world applications. Consequently, we employ TGP to reduce the feature scale, as illustrated in Fig. 2 (c), which significantly improves performance and enables practical deployment. Building on TGP, we propose simplified TGP, as shown in Fig. 2 (d), that merges feature interactions before and after pruning, achieving performance consistent with the original TGP while enhancing inference speed.

## 4.5 QUALITATIVE RESULTS

**Text-guided Pruning.** To visually demonstrate the process of our TGP, we visualize the results of two pruning phases, as shown in Fig. 4. In each example, the voxel features after scene-level pruning, the features after target-level pruning, and the features after target-level generative sparse convolution are displayed from top to bottom. It is evident that both pruning stages effectively achieve our intended effect: the scene-level pruning filters out the background and retained object voxels, and the target-level pruning preserves relevant and target objects. Moreover, during the feature upsampling process, the feature count nearly exponentially increases with the resolution enhancement due to generative upsampling. Without TGP, the voxel coverage would far exceed the range of the scene point cloud, which is unacceptable for inference. This also intuitively explains the significant impact of our TGP on both performance and inference speed.

**Completion-based Addition.** To clearly illustrate the function of our CBA, we visualize the adapt completion process in Fig. 5. The images below showcase several instances of excessive pruning. TGP performs pruning based on deep and low-resolution features, which can lead to excessive pruning, potentially removing entire or partial targets. This over-pruning is more likely to occur with small, as shown in Fig. 5 (a) and (c), narrow, as in Fig. 5 (b), or elongated targets, as in Fig. 5 (d). Based on this, our CBA effectively supplements the process using higher-resolution backbone features, thus dynamically integrating multi-level features.

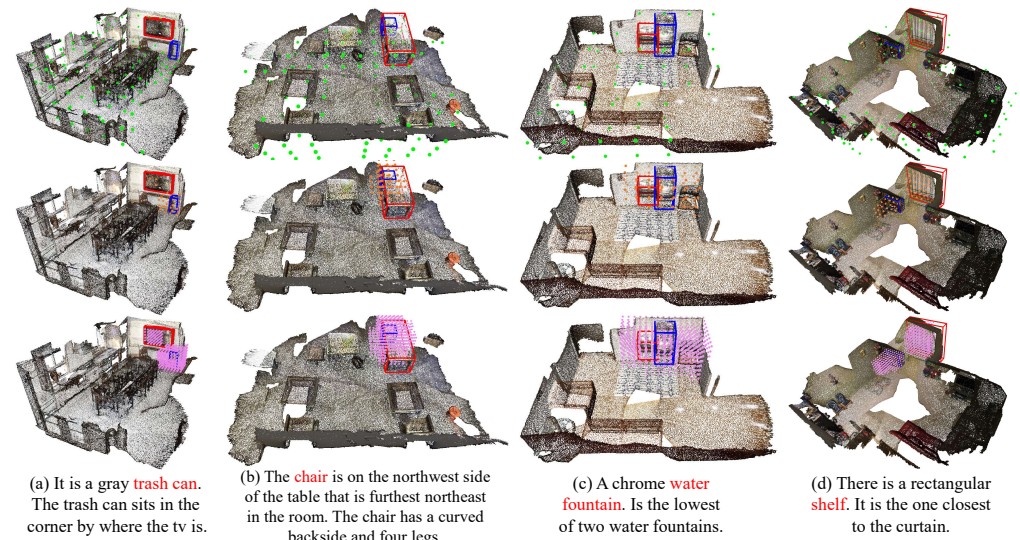

(a) It is a gray trash can. The trash can sits in the corner by where the tv is.

(b) The chair is on the northwest side of the table that is furthest northeast in the room. The chair has a curved backside and four legs.

(c) A chrome water fountain. Is the lowest of two water fountains.

(d) There is a rectangular shelf. It is the one closest to the curtain.

Figure 4: Visualization of the text-guided pruning process. In each example, the voxel features after scene-level TGP, target-level TGP and the last upsampling layer are presented from top to bottom. The blue boxes represent the ground truth of the target, and the red boxes denote the bounding boxes of relevant objects. ESS3D reduces the scale of features through two stages of pruning and progressively guides the network focusing towards the target.

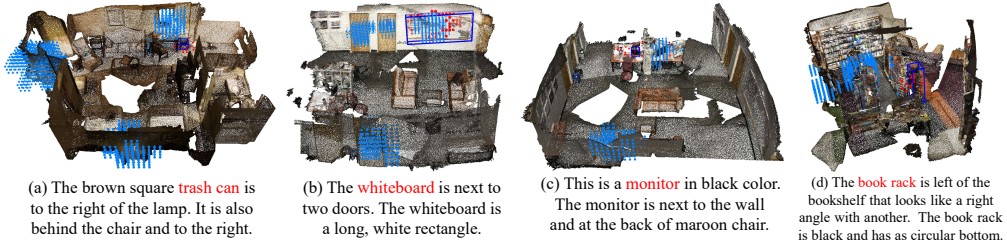

(a) The brown square trash can is to the right of the lamp. It is also behind the chair and to the right.

(b) The whiteboard is next to two doors. The whiteboard is a long, white rectangle.

(c) This is a monitor in black color. The monitor is next to the wall and at the back of maroon chair.

(d) The book rack is left of the bookshelf that looks like a right angle with another. The book rack is black and has as circular bottom.

Figure 5: Visualization of the completion-based addition process. The blue points represent the voxel features output by the target-level TGP, while the red points are the completion features predicted by the CBA. The blue boxes indicate the ground truth boxes. CBA adaptively supplements situations where excessive pruning has occurred.

## 5 CONCLUSION

In this paper, we present ESS3D, an efficient sparse single-stage method for real-time 3D visual grounding. Different architecture from previous 3D visual grounding (3DVG) methods, ESS3D builds on multi-level sparse convolutional architecture for efficient and fine-grained scene representation extraction. To enable the interaction between voxel and textual features, we propose text-guided pruning (TGP), which reduces the feature scale and guides the network to progressively focus on the target object. We further introduce completion-based addition (CBA) for adaptive multi-level feature fusion, effectively compensating for instances of over-pruning. Extensive experiments demonstrate the effectiveness of our proposed modules, resulting in an efficient 3DVG method that achieves state-of-the-art accuracy and inference speed.

**Potential Limitations.** Despite of the leading accuracy and inference speed, there are still some limitations of ESS3D. First, the speed of ESS3D is bit slower than ESS3D-B. Although ESS3D utilizes TGP to enable deep interaction between voxel and text features in an efficient way, it unavoidably introduces additional computational overhead compared with naive concatenation. In the future work, we aim to work on designing new operations for multi-modal feature interaction to replace the heavy cross-attention mechanism. Second, currently the input of 3DVG methods is a reconstructed point clouds. We will work on extending it to online setting with streaming RGB-D videos as input, which can support a wider range of practical application.

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

# A  APPENDIX

We provide detailed results for different subsets on ScanRefer (Chen et al., 2020), and qualitative comparisons of EDA (Wu et al., 2023) and ESS3D in the appendix.

## A.1  DETAILED RESULTS ON SCANREFER

To provide a detailed account of ESS3D's performance, we include the accuracy of ESS3D across various subsets of the ScanRefer dataset, as shown in Fig. 6. ESS3D achieves state-of-the-art accuracy, even when compared with two-stage methods, leading by $+1.13$ in Acc@0.5. In various subsets, ESS3D maintains comparable accuracy to both single-stage and two-stage state-of-the-art methods, while also demonstrating a level of efficiency that previous methods lack. Notably, the "multi-object" subset involves distinguishing the target object among numerous distractors of the same category within a more complex 3D scene. In this setting, ESS3D achieves a commendable performance of $42.37$ in Acc@0.5, further demonstrating that ESS3D enhances attention to the target object in complex environments through text-guided pruning and completion-based addition, enabling accurate predictions of both the location and shape of the target.

Table 6: Detailed Comparison of methods on the ScanRefer dataset evaluated at IoU thresholds of 0.25 and 0.5. ESS3D achieves state-of-the-art accuracy even compared with two-stage methods, with +1.13 lead on Acc@0.5. In various subsets, ESS3D achieves comparable accuracy to both single-stage and two-stage state-of-the-art methods. Additionally, ESS3D demonstrates a level of efficiency that previous methods lack.

| Method | Pipeline | Unique (~19%) | | Multiple (~81%) | | Accuracy | | Inference |
|---|---|---|---|---|---|---|---|---|
| | | 0.25 | 0.5 | 0.25 | 0.5 | 0.25 | 0.5 | Speed (FPS) |
| ScanRefer | Two-stage | 76.33 | 53.51 | 32.73 | 21.11 | 41.19 | 27.40 | 6.72 |
| TGNN | Two-stage | 68.61 | 56.80 | 29.84 | 23.18 | 37.37 | 29.70 | 3.19 |
| InstanceRefer | Two-stage | 77.45 | 66.83 | 31.27 | 24.77 | 40.23 | 30.15 | 2.33 |
| SAT | Two-stage | 73.21 | 50.83 | 37.64 | 25.16 | 44.54 | 30.14 | 4.34 |
| FFL-3DOG | Two-stage | 78.80 | 67.94 | 35.19 | 25.7 | 41.33 | 34.01 | Not release |
| 3D-SPS | Two-stage | 84.12 | 66.72 | 40.32 | 29.82 | 48.82 | 36.98 | 3.17 |
| BUTD-DETR | Two-stage | 82.88 | 64.98 | 44.73 | 33.97 | 50.42 | 38.60 | 3.33 |
| EDA | Two-stage | 85.76 | 68.57 | 49.13 | 37.64 | 54.59 | 42.26 | 3.34 |
| 3D-VisTA | Two-stage | 77.40 | 70.90 | 38.70 | 34.80 | 45.90 | 41.50 | 2.03 |
| VPP-Net | Two-stage | 86.05 | 67.09 | 50.32 | 39.03 | 55.65 | 43.29 | Not release |
| G$^3$-LQ | Two-stage | 88.09 | 72.73 | **51.48** | 40.80 | **56.90** | 45.58 | Not release |
| 3D-SPS | Single-stage | 81.63 | 64.77 | 39.48 | 29.61 | 47.65 | 36.43 | 5.38 |
| BUTD-DETR | Single-stage | 81.47 | 61.24 | 44.20 | 32.81 | 50.22 | 37.87 | 5.91 |
| EDA | Single-stage | 86.40 | 69.42 | 48.11 | 36.82 | 53.83 | 41.70 | 5.98 |
| G$^3$-LQ | Single-stage | **88.59** | **73.28** | 50.23 | 39.72 | 55.95 | 44.72 | Not release |
| ESS3D (Ours) | Single-stage | 87.25 | 71.41 | 51.04 | **42.37** | 56.45 | **46.71** | **12.43** |

## A.2 QUALITATIVE COMPARISONS

To qualitatively demonstrate the effectiveness of our proposed ESS3D, we visualize the 3DVG results of ESS3D alongside EDA (Wu et al., 2023) on the ScanRefer dataset (Chen et al., 2020). As shown in Fig. 6, the ground truth boxes are marked in blue, with the predicted boxes for EDA and ESS3D displayed in red and green, respectively. EDA encounters challenges in locating relevant objects, identifying categories, and distinguishing appearance and attributes, as illustrated in Fig. 6 (a), (c), and (d). In contrast, our ESS3D gradually focuses attention on the target and relevant objects under textual guidance and enhances resolution through multi-level feature fusion, showcasing commendable grounding capabilities. Furthermore, Fig. 6 (b) illustrates that ESS3D performs better with small or narrow targets, as our proposed completion-based addition can adaptively complete the target shape based on low-resolution feature maps.

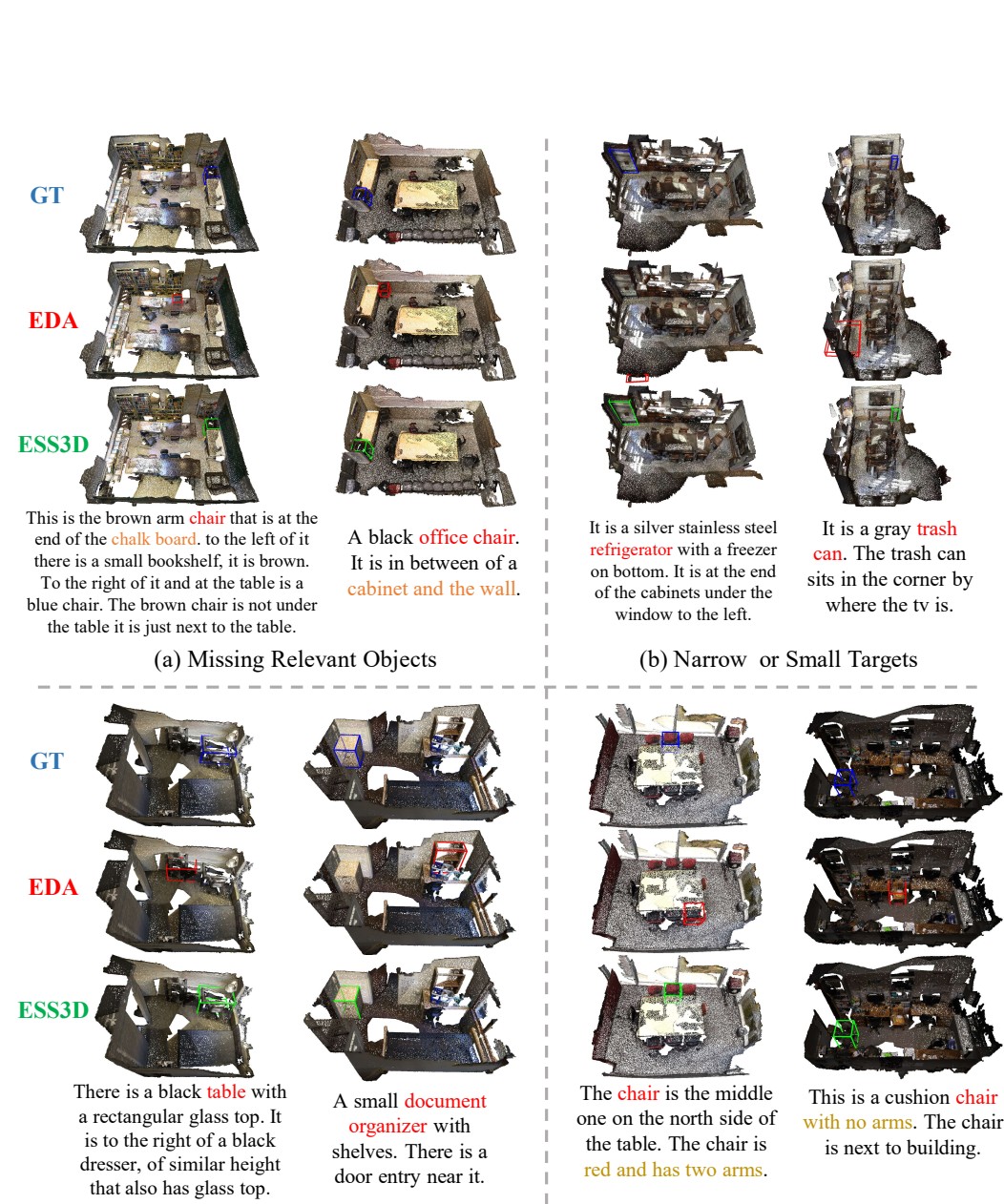

(a) Missing Relevant Objects

(b) Narrow or Small Targets

This is the brown arm chair that is at the end of the chalk board. to the left of it there is a small bookshelf, it is brown. To the right of it and at the table is a blue chair. The brown chair is not under the table it is just next to the table.

A black office chair. It is in between of a cabinet and the wall.

It is a silver stainless steel refrigerator with a freezer on bottom. It is at the end of the cabinets under the window to the left.

It is a gray trash can. The trash can sits in the corner by where the tv is.

(c) Category Error

(d) Appearance and Attributes

There is a black table with a rectangular glass top. It is to the right of a black dresser, of similar height that also has glass top.

A small document organizer with shelves. There is a door entry near it.

The chair is the middle one on the north side of the table. The chair is red and has two arms.

This is a cushion chair with no arms. The chair is next to building.

Figure 6: Qualitative results of EDA (Wu et al., 2023) and our ESS3D on ScanRefer dataset (Chen et al., 2020). In each description, the red annotations indicate the target object. The orange annotations in (a) refer to relevant objects, while the yellow annotations in (d) denote the appearance or attributes of the target. ESS3D demonstrates exceptional performance in locating relevant objects, narrow or small targets, identifying categories, and distinguishing appearance and attributes.

