# OpenReview forum: "Efficient Sparse Single-stage 3D Visual Grounding with Text-guided Pruning"
_ICLR.cc/2025/Conference — ICLR 2025 Conference Withdrawn Submission_

### Official Review · Reviewer_ZGs2 · 2024-10-30

**Soundness:** 2
**Presentation:** 2
**Contribution:** 3
**Rating:** 5
**Confidence:** 4

**Summary:**

The paper focuses on the efficiency problem of 3D Visual Grounding task. Instead of using point-based visual backbone, they adopt the convolutional one for efficiency. To efficiently fuse the visual feature with text feature, they propose text-guided pruning to sparsify the visual feature. To avoid over-pruning, they introduce completion-based addition to fix the over-pruned region.

**Strengths:**

1. The motivation is clear that current point-based SOTA 3DVG lack efficiency for inference. Therefore, it is meaningful to explore different 3d visual backbone. However, the paper lacks rigorous analysis to support this motivation.

2. The performance on ScanRefer, Nr3D and Sr3D are good.

3. The proposed method does not rely on object detector or ground truth bbox, which makes it meaningful and practical in real world.

**Weaknesses:**

1. **lack of comparison with low computational cost technique**: In line 60 to line 64, the authors claim that the cross-attention fusion in convolution-based  structure is high computational cost. However, several works in visual grounding field don't rely on cross-attention fusion, which might avoid this problem. For example, QRNet [1] uses Query-aware Dynamic Attention and VG-LAW [2] uses Language Adaptive Weight Generator. **Therefore, it is necessary to provide an efficiency comparison on those methods when transferred to convolutional-based structure, to evaluate if it is still a major problem regarding the multimodal fusion.**

2. **lack of rigorous analysis towards the bottleneck of efficiency**: The paper have mentioned several assumptions toward the inefficiency of current method, e.g. the use of furthest point sampling, the use of set abstraction, the use of object detector, the use of large transformer decoder. **However, the paper lacks a rigorous inference time anaylsis towards each of the assumptions,** and only provides an overall efficiency comparison. It is better to provide a breakdown of inference time for different components. What is the main bottleneck among all these mentioned problem? Can EDA / BUTD-DETR achieve a similar FPS with yours by simply reducing the layers of transformer encoder / decoder? If so, the motivation to use convolutional backbone might be weak.

3. **Writing needs to be improved**: The analysis of previous methods' efficiency should be put in another seperate section instead of the Method section. Readers would be confused if parts of Method are introducing other papers. For the method, the paper first provides a baseline structure in the "ESS3D-B" section, and then optimize it in the "Details of TGP" section, and then optimize it again in the "Simplification" section. Such an organization of writing is hard to follow and confusing. It's better to provide a seperate section to illustrate your path of exploration or organize it like ConvNets [3]. In the Method section, you can only clearly demonstrate your final version of the proposed model.

4. **Performance regarding efficiency is incremental**: Even though the paper claims to achieve +100% FPS improvement on the efficiency, the absolute FPS still remains in 12 FPS, which is far from the motivation for real-time (should be near 30 FPS).

5. **Some contradictory information**: The paper argue that aggressively downsampling of visual features would hurt performance but they also aggressively downsample the visual feature. The paper argue that furthest point sampling hurts efficiency but they also use furthest point sampling. Details refer to Question 1 & 2.

6. **Limited improvement from CBA**: As shown in Table 3, the CBA brings limited improvement. Specially, the +0.22 on Acc @ 0.5 is ​negligible.

[1] Shifting More Attention to Visual Backbone: Query-modulated Refinement Networks for End-to-End Visual Grounding, CVPR 2022

[2] Language Adaptive Weight Generation for Multi-task Visual Grounding, CVPR 2023

[3] A ConvNet for the 2020s, CVPR 2022

**Questions:**

1. The paper mentions that previous one-stage methods downsample the visual feature aggressively which might hurt the performance. However, the adopted MinkResBlocks in line 158 still downsamples the feature. And in line 214, the authors also aggressively prune the visual features. Therefore, it remains unclear about the comparison of the resolution / number of point feature between point cloud backbone and convolution backbone.

2. The authors claim that one of the reasons of the inefficiency of point-based backbone is the furthest point sampling, mentioned in line 144. However, the authors still use furthest point sampling mentioned in line 222 to line 223. Therefore, is furthest point sampling still a major problem for efficiency? If so, why the paper still explores it?

3. It is unclear how the authors perform concatenation with text feautures by duplicating in line 189 and Eq. (1), given that text features are in the shape of Length of tokens x channel dimension.

4. In the "Simplification" section, the authors remove some modules since they are computational cost. What exact cost are they? Could you provide quantitative results?

5. Do you use any pretrained parameters in your model? e.g. the visual backbone.

---

### Official Review · Reviewer_v5AX · 2024-11-02

**Soundness:** 3
**Presentation:** 3
**Contribution:** 3
**Rating:** 6
**Confidence:** 2

**Summary:**

This paper proposes to solve 3DVG more efficiently using single-stage sparse convolutional network. Two new modules are introduced: Text-Guided Pruning (TGP) and Completion-Based Addition (CBA). They help more efficiently fuse 3D scene and textual data by selectively pruning irrelevant regions and adaptively completing target information. The authors conducted experiments on the ScanRefer, Nr3D, and Sr3D datasets and showed their model’s superior accuracy and efficiency over existing methods.

**Strengths:**

The proposed designs and integration are well motivated and seem to work well. The model achieves leading accuracy and efficiency over existing methods.

**Weaknesses:**

-	While the introduction and integration of different components are well motivated and validated, the overall design is a little incremental.
-	Table 4 and the ablation study paragraph starting Line 442 are inconsistent. The authors stated that CBA at level 2 has negligible effects while CBA at level 1 is more impactful for improving accuracy. But Table 4 indicates the opposite: CBA at level 2 alone provides the best performance boost, much  bigger than CBA at level 1, and even more than having both level 1 and level 2.This confusing.
-	A minor visualization issue: The visualization of points in Figs. 4 and 5 blocked the objects in the scene, making them hard to see and verify. Making the superimposition half-transparent will help the visualization.

**Questions:**

Please address my concerns in the section of weaknesses.

---

### Official Review · Reviewer_N7XR · 2024-11-03

**Soundness:** 3
**Presentation:** 3
**Contribution:** 3
**Rating:** 8
**Confidence:** 3

**Summary:**

This paper proposes an one-stage framework for 3D visual grounding based on sparse convolution. To fully utilize the multi-level features, this paper introduces a text guided pruning strategy to prunthe unused features and further present completion-based addition to fix the over-pruned regions. On the ScanRefer and ReferIt3D datasets, it surpasses previsou fastest single-stage methods.

**Strengths:**

1. This paper is well written and easy to follow.
2. The experiments are comprehensive and the visualization results are beautiful.
3. The network performance is excellent.
4. The implementation details are complete.

**Weaknesses:**

1. The change of different levels of features is the main motivation of this work. The authors provide many examples to explain this.  In figure 4, I understand the scene-level and target-level TGP gradually prun the irrelevant features. To my understanding, the pink points ideally locate within the bounding boxes of the targets. But the visualization results seem different from this. It woule be better to provide much more detailed captions for these visualization results.

2. It would be better to list the network parameters.

Overall, this work presents clear motivation and comprehensive experiment analyses. It only requires a few minor improvements

**Questions:**

Please see the weekness

---

### Official Review · Reviewer_E7S5 · 2024-11-04

**Soundness:** 2
**Presentation:** 3
**Contribution:** 2
**Rating:** 5
**Confidence:** 4

**Summary:**

This paper proposes a single-stage framework for 3D visual grounding. In particular, it utilizes previous sparse convolutional network to utilize an Unet-like structure to interact point and text in multiple levels. A TGP module is designed to up-sample voxel features, and a CBA module is introduced to fuse multi-level features. Experiments are conducted on three common datasets.

**Strengths:**

1. This paper is easy to follow.

2. The designed multiple components are reasonable.

**Weaknesses:**

1. The motivation of using sparse convolution network is not clear. The authors simply claim that they utilize sparse convolution network as it performs well in other fields. This is not a convincing presentation and it weakens the novelty of this paper. The authors should provide the discussion of usage of sparse convolution network in depth, and discuss how sparse convolution network can alleviate the issues in existing 3DVG backbones.

2. About the novelty. The authors clarify that they propose several components to alleviate the issues of applying sparse convolution network into the 3DVG task. However, either the sparse convolution backbones nor the proposed attention-based components are not new. These modules are widely used in 2D or other 3D fields. Therefore, the novelty of the proposed framework is somewhat weak. The authors should provide detailed analysis and discussion.

3. Experiments are insufficient. First, there are many recent works summarized in paper [a]. The authors should carefully cite, compare, discuss with them. Second, a recent single-stage method [b] published in ECCV2024 achieves much better performance than this paper on Nr3D and Sr3D. Therefore, the claim of effectiveness of the proposed method is not convincing.

4. Why the sparse convolution network is efficient than other 3DVG backbones? It seems that the proposed method contains multiple components and is very complex. The authors should provide detailed complexity and efficiency analysis with existing 3DVG methods.

5. Ablation study is also insufficient. Since this paper proposes to utilize sparse convolution network to address the previous issues, the ablations on the architecture design of sparse convolution backbone is required, such as the ablation on layer number.

6. Global heatmap visualization on multi-level features should be also provided. Figure 4 is confusing.

[a] A Survey on Text-guided 3D Visual Grounding: Elements, Recent Advances, and Future Directions
[b] Multibranch collaborative learning network for 3d visual grounding

**Questions:**

1. The motivation of using sparse convolution network is not clear. The authors simply claim that they utilize sparse convolution network as it performs well in other fields. This is not a convincing presentation and it weakens the novelty of this paper. The authors should provide the discussion of usage of sparse convolution network in depth, and discuss how sparse convolution network can alleviate the issues in existing 3DVG backbones.

2. About the novelty. The authors clarify that they propose several components to alleviate the issues of applying sparse convolution network into the 3DVG task. However, either the sparse convolution backbones nor the proposed attention-based components are not new. These modules are widely used in 2D or other 3D fields. Therefore, the novelty of the proposed framework is somewhat weak. The authors should provide detailed analysis and discussion.

3. Experiments are insufficient. First, there are many recent works summarized in paper [a]. The authors should carefully cite, compare, discuss with them. Second, a recent single-stage method [b] published in ECCV2024 achieves much better performance than this paper on Nr3D and Sr3D. Therefore, the claim of effectiveness of the proposed method is not convincing.

4. Why the sparse convolution network is efficient than other 3DVG backbones? It seems that the proposed method contains multiple components and is very complex. The authors should provide detailed complexity and efficiency analysis with existing 3DVG methods.

5. Ablation study is also insufficient. Since this paper proposes to utilize sparse convolution network to address the previous issues, the ablations on the architecture design of sparse convolution backbone is required, such as the ablation on layer number.

6. Global heatmap visualization on multi-level features should be also provided. Figure 4 is confusing.

[a] A Survey on Text-guided 3D Visual Grounding: Elements, Recent Advances, and Future Directions
[b] Multibranch collaborative learning network for 3d visual grounding

---

### Note · Authors · 2024-11-15

I have read and agree with the venue's withdrawal policy on behalf of myself and my co-authors.